# GENERATING SEMANTIC ADVERSARIAL EXAMPLES WITH DIFFERENTIABLE RENDERING

## ABSTRACT

Machine learning (ML) algorithms, especially deep neural networks, have demonstrated success in several domains. However, several types of attacks have raised concerns about deploying ML in safety-critical domains, such as autonomous driving and security. An attacker perturbs a data point slightly in the concrete feature space (e.g., pixel space) and causes the ML algorithm to produce incorrect output ( a perturbed stop sign is classified as a yield sign). These perturbed data points are called adversarial examples, and there are numerous algorithms in the literature for constructing adversarial examples and defending against them. In this paper we explore *semantic adversarial examples (SAEs)* where an attacker creates perturbations in the semantic space representing the environment that produces input for the ML model. For example, an attacker can change the background of the image to be cloudier to cause misclassification. We present an algorithm for constructing SAEs that uses recent advances in differential rendering and inverse graphics. The generated SAEs are easily realizable in the physical world, and can be used to help the model adapt better to different domains.

## 1 INTRODUCTION

Machine learning (ML) techniques, especially Deep Neural Networks (DNNs), have been successful in several domains, such as finance and healthcare. However, several test-time (Biggio et al., 2013; Szegedy et al., 2014; Goodfellow et al., 2015; Kurakin et al., 2016) and training-time (Jagielski et al., 2018; Shafahi et al., 2018) attacks have made their adoption in high-assurance applications, such as autonomous driving and security, problematic. ML techniques, such as generative models, have also been used for nefarious purposes such as generating "deepfakes" (Liu et al., 2017; Zhu et al., 2017). Our focus in this paper is on test-time attacks in which an adversary generates a slightly perturbed sample to fool a classifier or an object-detector.

Let $X$ be the sample space and $Y$ be the space of labels. A classifier $F$ is a function from $X$ to $Y$. Given a sample $\mathbf{x} \in X$, most attacks for constructing adversarial examples find a perturbation $\delta$ with a small norm (typical norms that are used are $l_\infty$, $l_0$, and $l_2$) such that $\mathbf{x} + \delta$ has a different label than $\mathbf{x}$, *i.e.* $F(\mathbf{x}) \neq F(\mathbf{x} + \delta)$. In this paper we consider the problem of generating *semantic adversarial examples (SAEs)* (Hosseini & Poovendran, 2018; Joshi et al., 2019; Qiu et al., 2019; Dreossi et al., 2018b). In these examples, there is a richer set of transformations $\mathcal{T}$ that capture semantically-meaningful changes to inputs to the ML model. We assume a norm on $\mathcal{T}$ (this norm is induced by various parameters corresponding to the transformations, such as angle of rotation and size of the translation). In our universe, an adversary is given a sample $\mathbf{x}$ and wishes to find a transformation parameterized by $\theta \in \Theta$ with small norm such that $F(\tau(\mathbf{x}, \theta)) \neq F(\mathbf{x})$ (we consider untargeted attacks, but our ideas extend to targeted attacks as well)[1].

SAEs can also be viewed as outcomes of perturbations in a "rich" semantic feature space (*e.g.,*, texture of the image) rather than just the concrete feature space (*e.g.,*, pixels). Consequently, SAEs are physically realizable, and it is easy to understand how the changes in semantics results in an adversarial example. SAEs have been considered in the literature (Xiao et al., 2018; Dreossi et al., 2018b; Huang et al., 2019), but prior works typically consider a small set of fixed transformations (*e.g.,* rotation and translation, or modifying a single object's texture). Our goal is to flexibly support a

---

[1]$\tau$ is defined in § 3.2.

richer set of transformations implemented in a state of the art renderer (*e.g.,* changing the background of the image, weather conditions, or the time of day), as we specify in detail in § 2. There is evidence that SAEs can help with domain adaptation (Volpi et al., 2018) or making the control loop more robust (Dreossi et al., 2018b), further motivating our approach. We also plan to open source our code to advance research in this space.

To summarize, the main contributions of this paper are the following:

- We present a new class of test-time attacks in the form of SAEs. We demonstrate how to generate SAEs that support a rich set of transformations (refer § 3) using an inverse graphics framework (refer § 2). Specifically, we show how one can systematically take techniques to perform attacks in the pixel space such as FGSM (Goodfellow et al., 2015) and PGD (Madry et al., 2017) and transform them to their semantic counterparts.
- We evaluate the generated SAEs on the popular object detector SqueezeDet (Wu et al., 2016). By correctly choosing the semantic parameters, SAEs degrade performance (characterized by the mean average precision or mAP) by 28 percentage points (refer § 5.1).
- We show that by augmenting the dataset using SAEs, we can boost the robustness of SqueezeDet (characterized by mAP) by up to 15 percentage points (refer § 5.2).
- We show that SAEs generated using SqueezeDet do not transfer to YOLOv3 (Redmon & Farhadi, 2018), indicating a tight coupling between the model architecture and SAE generation (refer § 5.3).
- While augmentation with SAEs improves robustness against SAEs, augmentation using traditional pixel-based perturbations does not produce the same effect (refer § A.1).

## 2 RELATED WORK

**Adversarial Examples and Robustness:** There is extensive research for generating adversarial examples in the pixel space; we henceforth refer to these as *pixel-perturbations*. Goodfellow et al. (2015) propose the fast gradient sign method (FGSM) where inputs are modified in the direction of the gradients of the loss function with respect to input, causing a variety of models to misclassify their inputs. Madry et al. (2017) generalize this approach and propose the projected gradient descent (PGD) approach working using the same intuition. While these approaches suggest modifications to the raw pixel values, other methods of generating adversarial examples exist. Athalye et al. (2017) introduce an approach to generate 3D adversarial examples (over a chosen distribution of transformations). Engstrom et al. (2019) observe that modifying the spatial orientation of images results in misclassifications. Similarly, Geirhos et al. (2018) discovered that certain models are biased towards textural cues.

To improve robustness, current approaches include adversarial training (Madry et al., 2017), smoothing-based approaches (Cohen et al., 2019; Lécuyer et al., 2018), or through specific regularization (Raghunathan et al., 2018). An alternative approach, utilizing some notion of semantics, is advocated in the work of Guo et al. (2017). The authors augment the training set with transformed versions of training images, utilizing basic image transformations (*e.g.,*, scale and re-cropping) and total variance minimization, and demonstrate an improvement in robustness. Dreossi et al. (2018a) improve the robustness of SqueezeDet (Wu et al., 2016) through counterexample guided data augmentation; these counterexamples are synthetically generated by sampling from a space of transformations and applying them to original training images.

**Inverse Graphics:** The process of finding 3D scene parameters (geometric, textural, lighting, etc.) given images is referred to as inverse graphics (Baumgart, 1974). There is a history of using gradients to solve this problem (Blanz & Vetter, 2002; Shacked & Lischinski, 2001; Barron & Malik, 2015). Kulkarni et al. (2015) propose a model that learns interpretable representations of images (similar to image semantics), and show how these interpretations can be modified to produce changes in the input space. Pipelines for general differential rendering were proposed by Loper & Black (2014) and Kato et al. (2018). Li et al. (2018) design a general-purpose differentiable ray tracer; gradients can be computed with respect to arbitrary semantic parameters such as camera pose, scene geometry, materials, and lighting parameters. Yao et al. (2018) propose a pipeline that, through de-rendering obtains various forms of semantics, geometry, texture, and appearance, which can be rendered using a generative model.

**Semantic Adversarial Examples (SAEs)**: The notion of SAEs has been proposed in the literature in specific contexts. For example, in the NLP domain, Lei et al. (2018) propose a gradient guided greedy algorithm to make semantic changes to text documents. Sharif et al. (2016) propose attacks against facial recognition systems that are both inconspicuous and realizable. Similarly, Evtimov et al. (2017) obtain patches that can be added to road signs, rendering object detectors useless. In both these works, however, the modifications to semantics are not found by searching the space of semantic modifications (and found instead by *realizing* changes in pixel intensities); such approaches will fail to generalize for other domains. Dreossi et al. (2018b) present an approach to generate SAEs based on systematic sampling of the semantic space coupled with verification techniques for a closed-loop cyber-physical system containing the ML model, where they treat the ML model as a black box. However, this sampling-based "black box" approach faces scalability issues when dealing with high-dimensional spaces and also fails to exploit structure of the ML model to perform a targeted search for SAEs. Xiao et al. (2019) utilize a differentiable rendering framework to introduce changes in shape and texture that are capable of fooling a variety of ML models for various tasks. However, our approach takes into account advances in *differentiable inverse graphics* permitting us to change a richer set of semantic parameters. An interesting contribution by Qiu et al. (2019) suggests how a generative model can be used to introduce semantic modifications. However, generative models are notoriously difficult to train and operate, making them unusuable for a wide variety of tasks. Similar issues occur in the work by Joshi et al. (2019).

## 3 SEMANTIC ADVERSARIAL LEARNING

Consider a space $Z$ of the form $X \times Y$, where $X$ is the sample space and $Y$ is the set of labels. From here on we will assume that $X = \Re^n$. Let $H$ be a hypothesis space (*e.g.*, weights of a DNN). We assume a loss function $\ell : H \times Z \mapsto \mathbb{R}$ so that given a hypothesis $w \in H$ and a labeled data point $(x, y) \in Z$, the loss is $\ell(w, x, y)$. The output of the learning algorithm is a *classifier*, which is a function from $\Re^n$ to $Y$. To emphasize that a classifier depends on a hypothesis $w \in H$, which is output of the learning algorithm, we will denote it as $F_w$ (if $w$ is clear from the context, we will sometimes simply write $F$).

### 3.1 TRADITIONAL ADVERSARIAL EXAMPLES

We will focus our discussion on untargeted attacks, but our discussion also applies to targeted attacks. An adversary $A$'s goal is to take any input vector $\mathbf{x} \in \Re^n$ and produce a minimally altered version of $\mathbf{x}$, an *adversarial example* denoted by $A(\mathbf{x})$, that has the property of being misclassified by a classifier $F : \Re^n \rightarrow Y$. The adversary wishes to solve the following optimization problem:

$$\min_{\delta \in \Re^n} \quad \mu(\delta)$$
$$\text{such that} \quad F(\mathbf{x} + \delta) \neq F(\mathbf{x})$$

The various terms in the formulation are: $\mu$ is a norm on $\Re^n$; $\mu$ can be $l_\infty$, $l_0$, $l_1$, or $l_p$ ($p \geq 2$). While not representative, these norms are commonly used as an approximation of a human's visual perception (Sen et al., 2019). If $\delta$ is the solution of the optimization problem given above, then the adversarial example $A(\mathbf{x}) = \mathbf{x} + \delta$.

**FGSM.** The *fast gradient sign method (FGSM)* (Goodfellow et al., 2015) was one of the first untargeted attacks developed in literature. The adversary crafts an adversarial example for a given legitimate sample $\mathbf{x}$ by computing (and then adding) the following perturbation:

$$\delta = \varepsilon \operatorname{sign}(\nabla_{\mathbf{x}} L_F(\mathbf{x})) \tag{1}$$

The function $L_F(\mathbf{x})$ is a shorthand for $\ell(w, \mathbf{x}, l(\mathbf{x}))$, where $w$ is the hypothesis corresponding to the classifier $F$, $\mathbf{x}$ is the data point and $l(\mathbf{x})$ is the true label of $\mathbf{x}$ (essentially we evaluate the loss function at the hypothesis corresponding to the classifier). The gradient of the function $L_F$ is computed with respect to $\mathbf{x}$ using sample $\mathbf{x}$ and label $y = l(\mathbf{x})$ as inputs. Note that $\nabla_{\mathbf{x}} L_F(\mathbf{x})$ is an $n$-dimensional vector and $\operatorname{sign}(\nabla L_F(\mathbf{x}))$ is a $n$-dimensional vector whose $i^{th}$ element is the sign of the $\nabla_{\mathbf{x}} L_F(\mathbf{x})[i]$. The value of the *input variation parameter* $\varepsilon$ factoring the sign matrix controls the perturbation's amplitude. Increasing its value increases the likelihood of $A(\mathbf{x})$ being misclassified by the classifier $F$ but also makes adversarial examples easier to "detect" by humans. The key idea

is that FGSM takes a step *in the direction of the gradient of the loss function with respect to the input*, thus attempting to maximize the loss function using its first-order approximation. Recall that stochastic gradient descent (SGD) takes a step in the direction that is on expectation opposite to the gradient of the loss function because it is trying to minimize the loss function.

**PGD.** In *Projected Gradient Descent (PGD)* (Madry et al., 2017), we find a perturbation in an iterative manner. The PGD attack can be thought of an iterative version of FGSM. Assume that we are using the $l_p$ norm. Assume $\mathbf{x}_0$ is the original sample $\mathbf{x}$.

$$\mathbf{x}_{k+1} = \Pi_{B_p(\mathbf{x},\varepsilon)}(\mathbf{x}_k + \varepsilon \operatorname{sign}(\nabla_\mathbf{x} L_F(\mathbf{x}))) \tag{2}$$

The operator $\Pi_{B_p(\mathbf{x},\varepsilon)}(y)$ is the projection operator, *i.e.* it takes as input a point $y$ and outputs the closest point in the $\varepsilon$-ball (using the $l_p$-norm) around $\mathbf{x}$. The iteration stops after a certain number of steps (the exact number of steps is a hyperparameter).

## 3.2 SEMANTIC ADVERSARIAL EXAMPLES (SAEs)

Let $\mathcal{T} : (\Re^n \times \Theta) \to \Re^n$ be a set of transformations parameterized by a space $\Theta$, and $\mu$ is a norm over $\Theta$. The reader can think of $\Theta$ as parameters that control the transformations (*e.g.,* the angle of rotation). Given $\theta \in \Theta$, $\tau(x,\theta)$ is the image transformed according to the parameters $\theta$. We assume that there is a special identity element in $\Theta$ (which we call $\perp$) such that $\tau(\mathbf{x}, \perp) = \mathbf{x}$. An adversarial attack in this universe is characterized as follows:

$$\min_{\theta \in \Theta} \quad \mu(\theta)$$
$$\text{such that} \quad F(\tau(\mathbf{x},\theta)) \neq F(\mathbf{x})$$

In other words, we ideally want to find a "small perturbation" in the parameter space $\Theta$ that will misclassify the sample[2]. Consider the function $L_F(\tau(\mathbf{x},\theta))$. The derivative with respect to $\theta$ is $\left[\frac{\partial\tau}{\partial\theta}\right]^\top\Big|_\mathbf{x} \nabla_\mathbf{z} L_F(\mathbf{z})\big|_{\mathbf{z}=\tau(\mathbf{x},\theta)}$ (the notation $\left[\frac{\partial\tau}{\partial\theta}\right]^\top\Big|_\mathbf{x}$ is the transposed Jacobian matrix of $\tau$ as a vector-valued function of $\theta$, evaluated at $\mathbf{x}$, and $\nabla_\mathbf{z} L_F(\mathbf{z})\big|_{\mathbf{z}=\tau(\mathbf{x},\theta)}$ is the derivative evaluated at $\tau(\mathbf{x},\theta)$). The semantic version of FGSM (*sFGSM*) will produce the following $\theta$:

$$\theta^\star = \varepsilon \operatorname{sign}\left( \left[\frac{\partial\tau}{\partial\theta}\right]^\top\Big|_{(\mathbf{x},\perp)} \nabla_\mathbf{z} L_F(\mathbf{z})\big|_{\mathbf{z}=\tau(\mathbf{x},\perp)} \right) \tag{3}$$

The adversarial example $A(\mathbf{x})$ is $\tau(\mathbf{x},\theta^\star)$. Note that we do not assume any special properties about $\tau$, such as linearity. We only assume that $\tau$ is differentiable.

In a similar manner a semantic version of the PGD attack (*sPGD*) can be constructed. Let $\theta_0 = \perp$ and $\mathbf{x}_0 = \mathbf{x}$. The update steps correspond to the following two equations:

$$\theta_{k+1} = \Pi_{B_\mu(\theta_0,\varepsilon)}\left( \theta_k \oplus \varepsilon \operatorname{sign}\left( \left[\frac{\partial\tau}{\partial\theta}\right]^\top\Big|_{(\mathbf{x}_k,\theta_k)} \nabla_\mathbf{z} L_F(\mathbf{z})\big|_{\mathbf{z}=\tau(\mathbf{x}_k,\theta_k)} \right) \right)$$
$$\mathbf{x}_{k+1} = \tau(\mathbf{x}_0, \theta_{k+1})$$

Note that $\Pi_{B_\mu(\cdot,\cdot)}$ is the projection operator in the parameter space $\Theta$. We also assume that the projection operator will keep the parameters in the feasible set, which depends on the image (*e.g.,* translation does not take the car off the road). The operator $\oplus$ is the aggregation operator (similar to addition in $\Re^n$), but in the parameter space $\Theta$. The precise axioms satisfied by $\oplus$ depends on $\Theta$, but one axiom we require is:

$$\tau(\mathbf{x}, \theta_1 \oplus \theta_2) = \tau(\tau(\mathbf{x},\theta_1), \theta_2)$$

In fact, our recipe can be used to transform any attack algorithm such as Carlini & Wagner (2017) that adds a perturbation $\delta$ to its "semantic version" as follows:

---

[2]However, the existence of such a function for the semantic space is an open research question; the semantic parameter space is not homogeneous and it is unclear if one function can be used to capture the distance amongst all these transformations. Not only should the norm measure the changes in the semantic space, it should also approximate human perception. In its presence, one would be able to effectively measure the distortion introduced by the adversary in the semantic space.

- Replace $\delta$ with $\theta$.
- Replace $\mathbf{x} + \delta$ with $\tau(\mathbf{x}, \theta)$.
- Use chain rule to compute the gradients of terms that involve $\tau(\mathbf{x}, \theta)$.

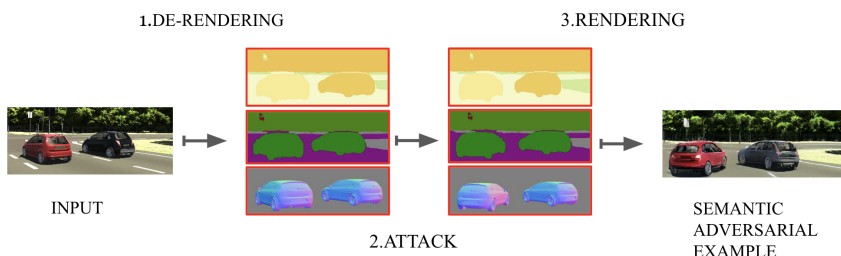

Figure 1: The input is de-rendered (step 1) to its intermediary representation (IR) - semantic, graphic, and textural maps. Then, this is adversarially perturbed (*e.g.*, the red car is rotated) as described in § 3.2 (step 2). The resulting IR is then re-rendered to the generate the SAE (step 3).

**Differentiable rendering and inverse graphics.** We apply the above framework to images by employing a differentiable renderer/de-renderer in an inverse graphics setting. Such an inverse graphics setting can be thought of two transformations: (a) a de-renderer $\beta : \Re^n \to \mathcal{S}$, and (b) a renderer $\gamma : \mathcal{S} \to \Re^n$. Here, $\mathcal{S}$ is the intermediate representation (IR). In the differentiable renderer/de-renderer we utilize (Yao et al., 2018), the IR contains a semantic map, texture codes, and 3D attributes. Let $\Theta$ be the set of changes to the IR (*e.g.,* change to the texture code to make it more cloudy) and $\perp \in \Theta$ corresponds to the identity. Suppose there is an operator $\alpha : (\mathcal{S} \times \Theta) \to \mathcal{S}$ that given a $\theta \in \Theta$ transforms the IR, *i.e.* $\alpha(s, \theta) = s'$ for $s, s' \in \mathcal{S}$. In this case, the function $\tau(\mathbf{x}, \theta)$ is equal to $\gamma(\alpha(\beta(\mathbf{x}), \theta))$. We use the fact that for differentiable renderers/de-renderers the functions $\beta, \gamma, \alpha$ are differentiable and hence attacks like sFGSM and sPGD can be implemented.

## 4 VALIDATION

In this section, we describe the various components used in our implementation to generate SAEs, and describe experiments carried out to determine the impact of choice of semantic parameters towards generating effective SAEs for object detectors. We stress that the exact choice of components are not important; our contribution is our automated approach to combine these components to generate SAES in a manner that is general (see § 3.2 for more details) and would work with any inverse graphics framework. We also believe that our methodology will work for other tasks such as classification and semantic segmentation; these tasks require optimizing over a different cost function. In other words, if there is an existing pixel attack, we should be able to easily transform it into a semantic attack (with the right inverse graphics framework).

### 4.1 IMPLEMENTATION DETAILS

The three main components required to successfully generate SAEs include: (a) a differentiable inverse graphics framework, (b) a victim model (which is also differentiable), and (c) an attack strategy. We describe our choices for the proof-of-concept below.

To obtain the semantics associated with our inputs and to generate the final SAEs, we use the inverse graphics framework(*i.e.* a combination of a semantic, textural and geometric de-rendering pipeline and a generative model for rendering) created by Yao et al. (2018). The models in this framework were trained entirely using the VKITTI dataset (Gaidon et al., 2016). These images comprise of simulations of cars in different road environments in virtual worlds. The VKITTI dataset is constructed using a novel real-to-virtual cloning methodology, mirroring many elements that exist in a real world. The de-rendering pipeline is used to obtain the initial semantic features associated with input images. These semantic features include (a) *color*: the car's texture codes, which change its color, (b) *weather*: the weather and time of day, (c) *foliage*: the surrounding foliage and scenery, (d) *rotate*: the car's orientation, (e) *translate*: the car's position in 2D, and (f) *mesh*: the 3D mesh which

provides structure to the car. Building an inverse graphics framework for other tasks/datasets is a non-trivial process which is outside the scope of this paper.

The final SAEs were produced using the generative model (which is part of the inverse graphics framework). Specific modifications were made to the differentiable graphics framework we used to ensure that gradients were easy to calculate. The codebase did not originally support end-to-end differentiation as each branch (semantic, geometric, textural) was trained separately. In particular, several image manipulation operations (normalization, rescaling through nearest-neighbor and bilinear interpolation) were implemented in a non-differentiable manner. We implemented the differentiable equivalents of these operations to allow backpropagation. Furthermore, we implemented a weak perspective projection for vehicle objects, as well as an improved heuristic for inpainting of gaps in the segmentation map due to object translations/rotations, in order to improve the quality of the rendering. Additionally, the inverse graphics framework and the object detector (which we describe next) had contrasting dependencies (for libraries, `tensorflow`, cuDNN, and CUDA); resolving these dependencies involved significant code rewriting.

We use the popular and representative SqueezeDet object detector (Wu et al., 2016) as the victim model. This model was originally trained on the KITTI dataset (Geiger et al., 2013). We perform transfer learning on this model using 6339 randomly chosen images from the VKITTI dataset; we wanted the object detector to better adapt to images outside the domain it was initially trained for. However, images produced by the differentiable graphics framework contain artifacts (*i.e.* distortions in the images); these artifacts could be mistaken for pixel perturbations and would impact our evaluation results. To deal with this issue, we retrain SqueezeDet using identity transform re-rendered images[3] produced by the generative model. Finally, we utilize these gradients and the semantics associated with each input in crafting adversarial attacks using the iterative sFGSM (for 6 iterations). On average, for a CPU-only implementation, generating such a SAE requires 42.81 seconds with about 85% of the time expended in the inverse graphics framework. We also stress that our choice of the number of iterations is restricted by our choice of the differentiable graphics framework. Using more iterations resulted in unintelligible outputs. We believe that the generation of SAEs is not inhibited by the complexity of the semantic parameter space; better engineering can substantially improve the outputs and the time it takes to generate them.

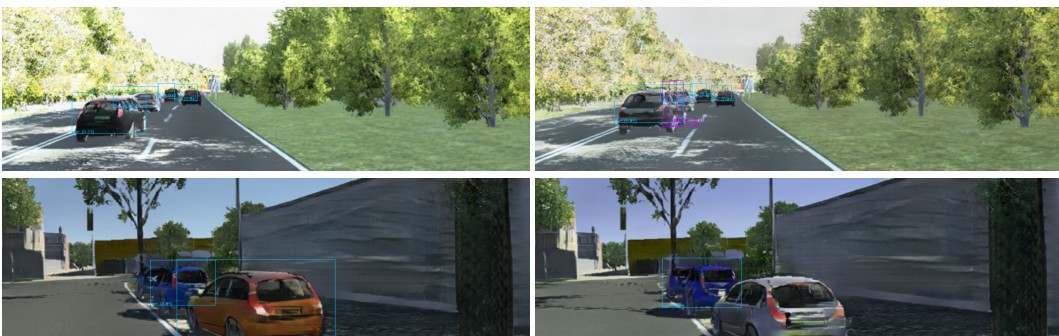

Figure 2: **Semantic space adversarial examples.** Benign re-rendered VKITTI image (left), adversarial examples generated by iterative sFGSM over a combination of semantic features (right). Cyan boxes indicate car detected, purple indicated pedestrian, and yellow indicate cyclist. The adversarial example introduces small changes in car positions and orientations, and noticeable changes in their color. This causes the network to detect pedestrians where there are none (top) and to fail to detect a car in the immediate foreground (bottom).

## 4.2 SELECTING SEMANTIC PARAMETERS

In the pixel perturbation setting, all pixels are equal *i.e. any* pixel can be perturbed. Whether such homogeneity naturally exists in the semantic space is unclear. However, we have additional flexibility; we can choose to modify any of the above listed semantic parameters independently without

---

[3]Images passed through the de-rendering and rendering framework, without modifying the IR or any associated semantic parameters.

altering the others, *i.e.* perform *single parameter modifications*. Alternatively, we can modify any subset of the parameters in unison, *i.e.* perform *multi-parameter modifications*. The degree of modification is determined by the *input variation/step-size* hyperparameter $\varepsilon \in [0, 1]$. In the context of pixel perturbations, the step-size corresponds to the maximum permissible change of a pixel. For SAEs, the value of $\varepsilon$ is proportional to the magnitude of the geometric and textural changes induced; the effect depends on the semantic parameter under consideration.

Large values of $\varepsilon$ result in unrealistic images created by the generative model (examples of this include perturbing the mesh to the point where cars are twisted into shapes no longer resembling vehicles). To avoid such issues and to simulate *realistic* transformations, we use a different step-size for each semantic parameter. We test various values of $\varepsilon$ for each semantic parameter, and report the best choice for brevity. Specifically, (a) *color*: $\varepsilon = 0.05$, (b) *weather*: $\varepsilon = 0.25$, (c) *foliage*: $\varepsilon = 0.10$, (d) *rotate*: $\varepsilon = 0.01$, (e) *translate*: $\varepsilon = 0.01$, and (f) *mesh*: $\varepsilon = 0.025$. We stress these hyperparameters were obtained after extensive visual inspection (by 3 viewers independently); norm-based approaches typically serve as a proxy for visual verification (Sen et al., 2019). Additionally, our choice in hyperparameters enables us to use the same ground truth labels throughout our experiments; *e.g.,* produced SAEs have bounding box coordinates that enable us to use the same ground truth labels as their benign counterparts[4]. We also measure the Fréchet Inception Distance (FID) scores (Heusel et al., 2017) for the generated SAEs; we observe that the score is 0.102362 (compared to 0.01651 for generating pixel perturbations), providing quantitative evidence that the SAEs are similar to the benign images they were generated from.

We produce 50 SAEs for each semantic parameter combination choice. We then evaluate the efficacy of generated SAEs on SqueezeDet by measuring its (a) recall percentage, and (b) mean average precision, or mAP, in percentage. These metrics have been used in earlier works (Xie et al., 2017).

| **Parameter** | *color* | *weather* | *foliage* | *translate* | *rotate* | *mesh* |
|---|---|---|---|---|---|---|
| **recall** | 100 | 100 | 100 | 100 | 100 | 98.7 |
| **mAP** | 99.5 | 98.8 | 99.7 | 99.2 | 98.2 | 98.7 |

Table 1: Performance of SqueezeDet on SAEs generated using single parameter modifications. The model had (a) recall = 100, and (b) mAP = 99.4 on benign/non-adversarial inputs. We observed that single parameter modifications are ineffective.

From Table 1, it is clear that single parameter modification is ineffective at generating SAEs. Thus, we generate SAEs using the multi-parameter modification method. To this end, we generated SAEs using the 57 remaining combinations of semantic parameters. One could consider a weighted combination of different semantic parameters based on a pre-defined notion of precedence. However, we choose a non-weighted combination. The results of our experiments are in Table 2. For brevity, we omit most of the combinations that do not result in significant performance degradation (and discuss the insight we gained from them in § 5.1). In the remainder of the paper, we report our evaluation using the *translate + rotate + mesh* parameter combination to generate SAEs.

| **Parameters** | *translate + rotate* | *translate + rotate + mesh* | *translate + mesh* | *rotate + mesh* |
|---|---|---|---|---|
| **recall** | 100 | 100 | 100 | 100 |
| **mAP** | 82 | 65.9 | 80.8 | 98.7 |

Table 2: Performance of SqueezeDet on SAEs generated using multi-parameter modifications. The model had (a) recall = 100, and (b) mAP = 99.4 on benign/non-adversarial inputs. We observed that certain combinations of multiple parameters are effective towards launching an attack.

---

[4]This fact is useful when we evaluate model robustness through retraining the models with SAEs as inputs, which we discuss in § 5.2

## 5 EVALUATION

We designed and carried out experiments to answer the following questions: (1) Do SAEs cause performance degradation in SqueezeDet?, (2) Can the generated SAEs be used for improving robustness?, and (3) Do the SAEs (generated using SqueezeDet) transfer to YOLOv3 (Redmon & Farhadi, 2018)?

We use 6339 images for training our SqueezeDet model, and evaluate the model using 882 SAEs. To evaluate the robustness, we augment the training dataset with 1547 SAEs and retrain the model. The various components of our framework and the datasets used are highlighted in § 4.1. Note that SqueezeDet's loss function comprises three terms corresponding to (a) bounding box regresson, (b) confidence score regression, and (c) classification loss. In our experiments, we target the confidence score regression loss term to impact the mAP and recall of the model. All code was written in `python`. Our experiments were performed on two servers. The SAE generation was carried out on a server with an NVIDIA Titan GP102 GPU, 8 CPU cores, and 15GB memory. All training and evaluation was carried out on a server with 264 GB memory, using NVIDIA's GeForce RTX 2080 GPUs and 48 CPU cores. Our experiments suggest that: (1) SAEs are indeed effective in degrading the performance of SqueezeDet. We also observe that the model is susceptible to changes that target the geometry of the input (cars in this case) rather than the changes in the background (refer § 5.1), (2) The generated SAEs do, in fact, help in improving model robustness. Our experiments show that SAE-based data augmentation can improve mAP by up to 15 percentage points (refer § 5.2), and (3) SAEs generated with SqueezeDet do not transfer to YOLOv3. (see § 5.3).

We do not report other metrics (classification accuracy, background error, etc.) associated with detection as our experiments are not designed to alter them.

### 5.1 EFFECTIVENESS OF SAES

The results in Table 2 in § 4.2 demonstrate the effectiveness of SAEs, and offer two insights.

First, the victim model was more susceptible to transformations that modify the geometry of the input (such as *translate* and *mesh*) than other types of transformations. This has dire implications for safety-critical applications; for the cars in our inputs, modifications in the *mesh* parameter results in *deformed* cars as outputs. These are common occurrences in sites of accidents, and need to be detected correctly. A combination of translations and rotations also seem to compound the degradation to the performance of the network (refer Table 2). This is most likely due to the introduction of unique angles and visual perspectives that are not frequently encountered in assembled datasets. Unlike pixel perturbations, SAEs are easy to interpret, *i.e.* we are able to understand how the model fails to generalize to specific changes in input semantics. Additionally, they are easier to realize *i.e.* the situations described above (related to translation and deformation of vehicles) occur on a daily basis. Intuitively, changing the geometry of the car can be viewed as targeting the perception of what a car really is – if the human can recognize that the object in question is a car but a model cannot, then the model is not exposed to the sufficient variety of car shapes, positions, and orientations that it may encounter in real-world scenarios; *i.e.* it is unable to domain adapt (Tzeng et al., 2017).

The second insight we gain is that the model was more susceptible to SAEs caused by changing multiple parameters simultaneously. We evaluate the model with 882 SAEs generated using a combination of the parameters listed in § 4.2. We observe that compared to the baseline performance on non-adversarial/benign inputs (recall = 93.63, mAP = 85.95), SAEs cause a significant performance degradation (recall = 93.17, mAP = 57.78). As stated before, these combinations are easily realizable, and the model's poor performance is indicative of poor domain adaptation.

### 5.2 DATA AUGMENTATION TO INCREASE ROBUSTNESS

As we have established that SAEs are effective in attacking SqueezeDet, we wished to enhance the model's robustness through data augmentation, as in Dreossi et al. (2018b). To this end, we carried out two sets of experiments. In the first, we incrementally (re)trained the benign SqueezeDet model on a combination of benign inputs and SAEs (4792+1547) for 24000 iterations. In the second, we tuned our benign model using just SAEs (1547) for 6000 iterations. The results of our experiments are presented in Table 3.

| Model | Baseline | Retrained (SAE + Benign) | Tuned (SAE) |
|---|---|---|---|
| **recall** | 93.17 | 92.97 | 92.15 |
| **mAP** | 57.78 | 72.76 | 72.63 |

Table 3: Performance of SqueezeDet on SAEs when (b) the model is retrained (on a combination of SAEs + benign inputs), and (c) the model is tuned (on just SAEs), compared to (a) the baseline model (trained on benign images) on SAEs. Both retraining and tuning improve mAP.

It is clear that both approaches provide comparable increase in mAP while not impacting recall. Additionally, we found that making a model robust to semantic perturbations through either procedure described earlier allowed us to achieve good performance on benign inputs. On benign inputs, we found that for the Retrained (SAE + Benign) model, recall = 93.7 and mAP = 84.73, while for Tuned (SAE), recall = 91.9 and mAP = 79.1. This is comparable to the performance of the baseline model (which was trained and validated on benign inputs), where recall = 93.6 and mAP = 86.17.

Our results suggests that SAE-based augmentation is a promising direction for exploration; based on insight from § 3, we could formulate a framework for semantic adversarial training, similar to (Madry et al., 2017). We leave the exact formulation to future work.

### 5.3 INVESTIGATING TRANSFERABILITY

To understand if SAEs transfer, we tuned a YOLOv3 network that was initially trained with the KITTI dataset, for 50 epochs. To do so, we use the same 6339 benign re-rendered images from the VKITTI dataset (as we did before), generated by the inverse graphics framework. The mAP of this network is 91.28%. When we tested the YOLOv3 network with the generated SAEs, we observed a drop in mAP to 87.50% (which is not as significant as in the case with SqueezeDet). This suggests that SAEs are effective against the network they are generated from, suggesting a tight coupling between the two. Similar observations have been made with pixel perturbations (Tramèr et al., 2017); further analysis is required to understand this phenomenon, which we hope to pursue in future work.

While we investigate if SAEs transfer across different models and architectures, an interesting question revolves around the transferability of SAEs across tasks. We believe that SAEs generated to fool object detectors will not work against classification or semantic segmentation networks as the generation process was not targeted for these tasks. That being said, it is trivial to modify our general construction (by changing the cost/loss function) and generate SAEs for tasks such as classification and semantic segmentation. We leave generating such SAEs for future work.

## 6 CONCLUSIONS

In this paper, we describe semantic adversarial examples (SAEs), where adversaries perturb the semantics of inputs to produce outputs that are misclassified. Such instances are easier to realize in the physical world, and are more interpretable than their traditional pixel-based counterparts. We propose an algorithm to construct SAEs using advances in differentiable rendering, and evaluate the effectiveness of our approach. We observe that SAEs cause performance degradation in object detector networks (SqueezeDet), and that data augmentation using SAEs increases robustness of the model.

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

# A  APPENDIX

## A.1  PIXEL PERTURBATIONS VS. SAES

Our experiments suggested that pixel perturbations are more effective in degrading SqueezeDet's performance (recall=2.9, mAP=0.05); we conjecture this is due to the larger feature space within which a solution for the optimization can be found *i.e.* the space of pixels is larger than the space of specific semantic parameters/transformations we consider.

To measure the robustness provided by pixel perturbations against semantic perturbations, we performed the same experiment as in § 5.2. In one experiment, we retrained the benign SqueezeDet model on a combination of benign inputs and pixel perturbations (4792+1547) for 24000 iterations. In another experiment, we tuned our model using just pixel perturbations (1547) for 6000 iterations. The results of our experiments are presented in Table 4.

| Model | Retrained (Pixel + Benign) | Tuned (Pixel) |
|---|---|---|
| **recall** | 92.4 | 90.43 |
| **mAP** | 56.7 | 55.35 |

Table 4: Performance of SqueezeDet on SAEs when the model is (a) retrained (on pixel perturbations + benign inputs), and (b) tuned (on just pixel perturbations). Retraining and tuning (with pixel perturbations) is ineffective against SAEs.

We observed that data augmentation using pixel perturbations *does not* increase the robustness to SAEs. Pixel perturbations are more general, and do not capture the effects induced by SAEs. Consequently, we wished to understand if we could get the best of both worlds *i.e.* robustness against both pixel perturbations and SAEs. We report the results of this experiment in Appendix A.

## A.2  HYBRID TUNING

To understand if we are able to obtain robustness against both pixel perturbations and SAEs, we repeated our experiments from earlier sections (§ 5.2); this time, we tuned the *model robust to SAEs* (obtained after 6000 iterations) using pixel perturbations for 6000 iterations - denoted Tuned (Pixel); we also tuned the *model robust to pixel perturbations* (obtained after 6000 iterations) using SAEs for 6000 iterations - denoted Tuned (SAE).

| Model | Tuned (SAE) | Tuned (Pixel) |
|---|---|---|
| **recall** | 93.2 | 91.7 |
| **mAP** | 73.8 | 65.7 |

(a) Validation set used: SAEs

| Model | Tuned (SAE) | Tuned (Pixel) |
|---|---|---|
| **recall** | 29.5 | 90.6 |
| **mAP** | 7.65 | 75.15 |

(b) Validation set used: pixel perturbations

Table 5: Performance of dual tuning (*i.e.* tuning a model robust to SAEs with pixel perturbations, and tuning a model robust to pixel perturbations with SAEs). The generality of pixel perturbations is masked by the specificity of SAEs (Table 5b - column 1).

We validated both the generated models against (a) SAEs, and (b) pixel perturbations. Our results are presented in Table 5. While the order of tuning did not greatly impact robustness in the face of SAEs, it had a direct impact for pixel perturbations (as evident from Table 5b). We explain the observation in Table 5b using manifold learning. The detector is assumed to generalize over unobserved data on the data manifold. Due to the limited amount of sampling, the detector fails to generalize to SAEs lying on these uncovered regions; learning them aids generalization. Pixel perturbations, however, do not lie on the data manifold (due to its larger degree of freedom). Tuning the model using pixel perturbation is analogous to learning a thickened manifold (with the same uncovered regions as before). Now, if we tune this model, the detector learns uncovered regions in a thickened (and incorrect) manifold; this impacts generalization, further making it susceptible to

pixel perturbations. Another intuition could be based on the generality of pixel perturbations; tuning first on such general perturbations impacts generalization (Tsipras et al., 2018). When such a model is tuned on SAEs, we gain generalization at the expense of robustness.

