# OpenReview forum: "Generating Semantic Adversarial Examples with Differentiable Rendering"
_ICLR.cc/2020/Conference — Reject_

### Official Review · AnonReviewer3 · 2019-10-20
**Official Blind Review #3**

**Rating:** 3

**Review:**

This paper proposes to generate "semantic adversarial examples" (SAEs) by applying small perturbations in the semantic space, by contrast to more standard adversarial examples that apply perturbations to the image intensities. The concept of SAE has already  been considered in a few papers (where for example the color distribution of the images are changed).

What the paper proposes is to consider small perturbations of the parameters of a scene. For example, a car can be slighted rotated in 3D, or its color changed, without changing its 2D bounding box. If the perturbation is differentiable, and with a differentiable rendering, it is possible to generate SAEs with optimization algorithms that generalize algorithms developed for "classical" adversarial examples (FGSM and PGD).
In practice the perturbations are performed by using the method from Yao et al. "3D-Aware Scene Manipulation via Inverse Graphics" (NIPS'18).

The paper shows that their SAEs disturbed an object detection method (SqueezeDet), and that training SqueezeDet with SAEs improves its robustness to their SAEs.

I found the paper interesting, however I am not convinced by the general concept:
* While the method from Yao et al is supposed to work on real images, it requires to detect and estimate the pose and the shape of 3D objects, which is still challenging to do on real images. This is probably why the paper considers only synthetic images (VKITTI), to be able to generate a large number of SAEs.

* The generated images are also valid images, in the sense that for a slightly different scene, they would have been generated by the image renderer used for VKITTI. It sounds more like what the authors have is a method that can generate scenes to improve the performance of an object detector when trained on synthetic images, rather than a method  to be more robust to malicious attacks. This is very interesting, however the paper is not branded toward this  application.

**Experience Assessment:**

I have read many papers in this area.

**Review Assessment: Checking Correctness Of Derivations And Theory:**

I assessed the sensibility of the derivations and theory.

**Review Assessment: Checking Correctness Of Experiments:**

I carefully checked the experiments.

**Review Assessment: Thoroughness In Paper Reading:**

I read the paper thoroughly.

---

> ### Author Response · Authors · 2019-11-14
> **Thank you for your valuable feedback!**
>
> We thank the reviewer for his/her insightful comments. We provide clarifications for some of the important questions raised.
>
> 1. Difficulties with inverse graphics: We thank the reviewer for pointing out difficulties in training/using the inverse graphics pipeline. We stress that our contribution is not in tuning/testing/training the inverse graphics pipeline (as we state in the paper in Section 4.1), but in using a pre-trained inverse graphics pipeline to generate SAEs. While pose estimation is a challenging problem in other datasets, and designing a framework to capture pose is a hard problem, such a problem (and its solution) is orthogonal to our work.
> 2. Use-Cases: We stress that our applications are beyond malicious attacks, and aids in domain adaptation (in the introduction and the abstract). We re-iterate this message through the revised version of the paper submitted. We thank the reviewer for their feedback in improving the motivation and message of the paper.

---

### Official Review · AnonReviewer1 · 2019-10-24
**Official Blind Review #1**

**Rating:** 3

**Review:**

Review: This paper focuses on generating adversarial perturbation in semantic space. The main contribution of this paper is to propose a general method to generate semantic adversarial examples by using advances in differential rendering and inverse graphics.

Pros:
1. Generally, the presentation is clear and easy to follow.
2. A general way to transform any pixel-attack algorithm to its “semantic version” is novel.

Cons:
1. There are already a lot of ways to make adversarial examples even semantic adversarial examples. It’s not enough to just propose a new way to make adversarial examples. I think this paper would be more compelling if proposed SAEs have some special property (e.g, easy to take effect in the real world or stronger robustness against defense strategy).
2. In section 5.3, the authors show that data augmentation using SAEs increase the robustness to SAEs, but pixel perturbation AEs do not. This result is of little value. It is obvious that data augmentation using SAEs can increase more robustness to SAEs or pixel-perturbation AEs can increase more robustness to pixel-perturbation AEs. The authors are suggested to compare changes in general robustness caused by two types of data augmentations. For example, compare minimum adversarial distortion.

**Experience Assessment:**

I have published one or two papers in this area.

**Review Assessment: Checking Correctness Of Derivations And Theory:**

I assessed the sensibility of the derivations and theory.

**Review Assessment: Checking Correctness Of Experiments:**

I assessed the sensibility of the experiments.

**Review Assessment: Thoroughness In Paper Reading:**

I read the paper at least twice and used my best judgement in assessing the paper.

---

> ### Author Response · Authors · 2019-11-14
> **Thank you for your valuable feedback!**
>
> We thank the reviewer for his/her valuable feedback. Below, we respond to some of the questions raised by the reviewer.
>
> 1. Motivation + Novelty: While we believe that pixel perturbations are hard to realize in the physical world, the SAEs we generate are much easier to realize. Thanks to your feedback, we re-iterate this point several times through the paper to make it clearer to the reader. We also compare and contrast our approach of generating SAEs with prior works in the space (see the text in blue in the related work section); the generality of our approach coupled with the ability to easily transform existing pixel perturbation attacks to attacks capable of generating SAEs is novel and has not been studied before.
> 2. Results: We have moved the results regarding the impact of SA-training on PP and vice-versa to the appendix (now Appendix A1) to make for clearer reading.
> 3. Adversarial Distortion: We believe that measuring the distortion caused by the adversary is a challenging proposition in the context of SAEs; in the case of pixel perturbations, adversarial distortion is measured using p-norms, which serve as a proxy for visual perception. However, there exists no function that accurately captures changes in semantics, and ties it to visual perception. Should such a function exist, we will be able to easily add it to our optimization framework, and produce SAEs with lesser computational overhead. We thank the reviewer for pointing this out; we have clarified this in the paper (added as a footnote on page 4), and pose discovering such a function as an open question to researchers in the community.

---

### Official Review · AnonReviewer2 · 2019-10-24
**Official Blind Review #2**

**Rating:** 3

**Review:**

This paper studies the problem of semantic adversarial attacks with a differentiable de-rendering and rendering pipeline. More specifically, this paper proposed a variant of FGSM (Goodfellow et al. 2015) and PGD (Madry et al. 2017) by extending the traditional Lp-bounded adversarial attacks in the rich semantic space. It considered a list of semantic parameters including color, weather, foliage, rotation, transformation, and object shape. To facilitate back-propagation and improve the quality of rendering, this paper re-implemented the differentiable equivalents of several image manipulation operations based on the previous work (Yao et al. 2018). For experimental evaluations, this paper selected the object detector SqueezeDet (We et al. 2016) as the target model for attack on the virtual KITTI dataset. Experiments demonstrated that the generated semantic adversarial examples (SAEs) can attack the SqueezeDet (see Table 1 and Table 2). By re-training with augmented data by the proposed method, the robustness of SqueezeDet (see Table 3) can be further improved.

Overall, this is an okay paper with incremental technical novelty and clear presentation. Reviewer has several concerns regarding the experiments.

(1) This paper only conducts experiments on virtual KITTI dataset, a synthetic benchmark for object detection and semantic segmentation. In general, studying the adversarial examples in the synthetic domain seems not a significant contribution. Reviewer would like to know the performance on the real dataset such as Cityscape (used in Yao et al. 2018) and other challenging indoor datasets such as ADE20K. At least, reviewer would like to know whether re-training on adversarial examples help to improve the performance on real dataset?

(2) The conclusion is largely based on the 1547 semantic adversarial examples generated, while there are more than 4K synthetic images in the dataset. This seems contradicts against the flexibility of generating semantic adversarial examples described in the paper (e.g., single parameter modifications). Reviewer suspects the proposed differentiable rendering pipeline is not very effective so that generating SAEs requires quite a bit exhaustive search over the parameter space. Please comment on the average running time for generating a semantic adversarial example. How does that compare to generating a pixel-based adversarial example?

(3) While several different quantitative analyses have been conducted, this paper only provides two examples as the qualitative result (see Figure 2). It would be more convincing if this paper provides more such examples in the appendix. In addition, ablation studies on semantic parameters are currently missing. Furthermore, reviewer wonders if it is possible to report the FID score and make sure the generated adversarial examples have the same distribution as ground-truth images.

(4) SqueezeDet is the only model used in the paper. Please also consider other models and report the attack performance and transferability. In a high-level, reviewer would like to know whether the proposed differentiable rendering method generalizes to other tasks including semantic segmentation and depth prediction.

(5) The following paper is related (see Figure 5 of MeshAdv paper), but not even mentioned here. Reviewer would like to see the comparison between the proposed method and the MeshAdv baseline.

-- MeshAdv: Adversarial Meshes for Visual Recognition. Xiao et al. In CVPR 2019.


**Experience Assessment:**

I have published one or two papers in this area.

**Review Assessment: Checking Correctness Of Derivations And Theory:**

I carefully checked the derivations and theory.

**Review Assessment: Checking Correctness Of Experiments:**

I carefully checked the experiments.

**Review Assessment: Thoroughness In Paper Reading:**

I read the paper thoroughly.

---

> ### Author Response · Authors · 2019-11-14
> **Thanks for your valuable feedback!**
>
> We thank the reviewer for his/her insightful comments. We were able to carry out some of the experiments suggested, and present our responses to the important questions raised.
>
> 1. Dataset: We chose the VKITTI dataset for our evaluation as the inverse graphics framework was trained on the VKITTI dataset. Retraining the inverse graphics framework on a different dataset is a difficult proposition, and one that was beyond the scope of our work. Our work suggests that if there exists an inverse graphics framework (suitably trained on the appropriate dataset, be it real or synthetic), then our proposed framework can generate semantic adversarial examples (SAEs). As suggested by the reviewer, we believe that it is important to vary the dataset to prove generality of the approach. However, there has been no prior work in adversarial example literature that suggests that some datasets have “hard to find/generate adversarial examples” (or are more challenging than others);  the difficulty of generating adversarial examples is more closely tied to the model used to generate the adversarial examples.
> 2. Exhaustive search over parameter space: We believe that generating SAEs requires no more/less search over the parameter space than in the conventional, pixel perturbation (PP) setting. In fact, the semantic feature space is usually much lower-dimensional than the concrete, pixel space. That being said, the time it takes to generate SAEs is larger than generating PPs; this again, is not a limitation of our approach, but instead due to inefficiencies in the renderer implementation (which we highlighted in Section 4, but will expand the discussion further).
> 3. FID Scores/Realism: We conduct the evaluation with FID scores using code from https://github.com/mseitzer/pytorch-fid; our analysis indicates that the FID score is 0.10632; this low value suggests that the generated SAEs lie in distribution. We thank the reviewer for pointing this out, and have incorporated the same in Section 4.2 in the paper (text marked in blue).
> 4. Model Transferability: We thank the reviewer again for providing us an opportunity to enhance our evaluation; we carried out tests to measure transferability by observing the degradation caused by the SAEs generated using SqueezeDet on the YoloV3 object detection network (which was again tuned with the inverse graphics re-rendered benign VKITTI image). We observe that the generated SAEs do not transfer. These results have also been added to Section 5.3 in the revised draft uploaded (text in blue). We believe that more in-depth analysis is required to understanding transferability, and hope to carry this out in future work.
> 5. Task Transferability: We believe that the generated SAEs will not be adversarial for different tasks (such as classification/semantic segmentation). Again, this is not a limitation of our approach. By changing the loss function we optimize over, one can easily generate SAEs for classification/semantic segmentation using the general approach we provide.
> 6. Related Work: We compare and contrast our work with related work (such as MeshAdv, which we now cite) in the modified related work section (text in blue). In summary, our approach contains a much broader set of transformations/semantics than considered by our contemporaries.  Our approach also suggests that with a suitably trained inverse graphics pipeline, off-the-shelf attacks (to generate pixel perturbations) can be easily transformed to generate SAEs.

---

### Decision · Program_Chairs · 2019-12-19

**Decision:**

Reject

**Comment:**

The authors present a way for generating adversarial examples using discrete perturbations, i.e., perturbations that, unlike pixel ones, carry some semantics. Thus, in order to do so, they assume the existence of an inverse graphics framework. Results are conducted in the VKITTI dataset. Overall, the main serious concern expressed by the reviewers has to do with the general applicability of this method, since it requires an inverse graphics framework, which all-in-all is not a trivial task, so it is not clear how such a method would scale to more “real” datasets. A secondary concern has to do with the fact that the proposed method seems to be mostly a way to perform semantic data-augmentation rather than a way to avoid malicious attacks. In the latter case, we would want to know something about the generality of this method (e.g., what happens a model is trained for this attacks but then a more pixel-based attack is applied). As such, I do not believe that this submission is ready for publication at ICLR. However, the technique is an interesting idea it would be interesting if a later submission would provide empirical evidence about/investigate the generality of this idea.